# Calcium- and Integrin-Binding Protein 2 (CIB2) in Physiology and Disease: Bright and Dark Sides

**DOI:** 10.3390/ijms23073552

**Published:** 2022-03-24

**Authors:** Giuditta Dal Cortivo, Daniele Dell’Orco

**Affiliations:** Department of Neurosciences, Biomedicine and Movement Sciences, Section of Biological Chemistry, University of Verona, 37134 Verona, Italy; giuditta.dalcortivo@univr.it

**Keywords:** non-syndromic deafness, Usher syndrome, hearing, integrin signaling, mechanoelectrical transduction

## Abstract

Calcium- and integrin-binding protein 2 (CIB2) is a small EF-hand protein capable of binding Mg^2+^ and Ca^2+^ ions. While its biological function remains largely unclear, an increasing number of studies have shown that CIB2 is an essential component of the mechano-transduction machinery that operates in cochlear hair cells. Mutations in the gene encoding CIB2 have been associated with non-syndromic deafness. In addition to playing an important role in the physiology of hearing, CIB2 has been implicated in a multitude of very different processes, ranging from integrin signaling in platelets and skeletal muscle to autophagy, suggesting extensive functional plasticity. In this review, we summarize the current understanding of biochemical and biophysical properties of CIB2 and the biological roles that have been proposed for the protein in a variety of processes. We also highlight the many molecular aspects that remain unclarified and deserve further investigation.

## 1. Introduction

More than two decades ago, Seki et al. isolated from a cDNA library of human fetal brains a clone encoding a 187 amino acid protein, initially named kinase interacting protein 2 and currently referred to as calcium-and integrin-binding protein 2 (CIB2) [1]. Examination of the transcript highlighted a broad distribution in various human tissues, including brain (both fetal and adult), heart, kidney, lung, thymus, spleen, placenta, ovary, and testis, thus suggesting that the product of the *CIB2* gene is involved in basic cellular functions [1]. Further studies also confirmed a broad expression of CIB2 in mice, specifically in the inner ear, outer and inner retina, retinal pigmented epithelium [2], and skeletal muscle, with higher expression levels in sarcolemma, the myotendinous junction, and the neuromuscular junction in adult mice and lower expression in brain and lungs [3]. Evidence of CIB2 expression has also been reported in adult rat brain, with specific localization in the hippocampus and in the sensory, entorhinal, and prefrontal cortex, with significant intracellular localization at the Golgi apparatus and neurites [4]. High expression levels of the ortholog CIB2 in sheep have been found by RT-PCR in different tissues, mainly stomach, heart, and ovary [5].

The extremely wide expression of CIB2 in various tissues of different organisms suggests its implication in a wide variety of biochemical processes. A broad and multifaceted biological function is consistent with its membership in the CIB family of Ca^2+^- and Mg^2+^-binding proteins [6]. This family includes homologous proteins showing evolutionary relationships with the class of neuronal calcium sensors (NCS) [7]. The biological role of CIB2 was initially associated with its interaction with specific integrins involved in intra- and extracellular signaling pathways [3,8]. More recently, however, increasing lines of evidence have associated CIB2 with hearing physiology and pathology [9], suggesting that CIB2 is an essential component for the normal development of hair cells in the inner ear and possibly a structural component of the hair cell mechanotransduction complex [10,11]. In parallel, biochemical and biophysical studies [12,13,14] have highlighted similarities but also important differences between CIB2 and other members of the CIB family, which should be considered when investigating its versatile and still largely unknown functions and involvement in processes as diverse as autophagy [15], cancer [16], and muscular dystrophy [3].

In this review, we summarize the current knowledge on the biochemical and biophysical properties of CIB2 and its putative mechanisms of action in the multitude of physiological and pathological processes where its involvement has been postulated or demonstrated. We also highlight the many issues that need to be clarified in future studies in order to reach a definitive statement on the role of CIB2 in health and disease.

## 2. Structural, Biochemical, and Biophysical Properties of CIB2

The discovery of the *CIB* gene family dates back to the 1990s, with the identification of the first member named calcium- and integrin-binding protein (CIB1) [17] or KIP (kinase interacting protein) [18] based on the putative function, which showed 58% sequence similarity with calcineurin B and 56% with calmodulin. Since then, several structural studies [13,19,20] have clarified the features shared by the CIB family members, which contain three further homologs, *CIB2*, *CIB3*, and *CIB4*, in the human genome, and highlighted commonalities with the neuronal calcium sensor proteins (NCS) family.

Human *CIB2* (NG_033006) is a 33,929 bp gene located on chromosome 15 (15q25.1) encoding four different isoforms, consisting of 4–6 exons [2]. The canonical sequence (Uniprot code: O75838-1) encodes a 187 amino acid protein of 21.6 kDa harboring two functional EF-hands, the metal-binding helix–loop–helix motif characterizing the largest family of Ca^2+^-binding proteins in eukaryotic cells [21]. Functional EF-hands are located at the C-terminal domain of CIB2 and will be referred to as EF3 and EF4 in this review to distinguish them from the non-functional EF1 and EF2 located in the N-terminal domain (Figure 1). The three-dimensional structure of CIB2 is currently unknown, although homology models have been built based on the experimental structure of CIB1 [14] and CIB3 [20].

### 2.1. CIB2 Has the Potential to Work as a Ca^2+^ and Mg^2+^ Sensor Protein

Since its discovery and based on the homology with CIB1, CIB2 was expected to act as a Ca^2+^ sensor, thereby changing its conformation upon the binding of Ca^2+^ ions and possibly other cations and acquiring a structure that allows the regulation of specific molecular targets. Human CIB2 was, therefore, heterologously expressed in *E. coli* and purified to study its structural and biophysical properties. Huang et al. confirmed that CIB2 acts as a typical EF-hand calcium sensor, which undergoes a significant Ca^2+^ induced conformational change, thereby exposing a hydrophobic patch and increasing the content of secondary structure—essentially α-helix—as proven by far-UV circular dichroism (CD) spectroscopy [13]. Interestingly, the protein was also shown to respond to Mg^2+^ with similar intensity, although 8-anilinonaphthalene-1-sulfonic acid (ANS) fluorescence showed that the most prominent conformational change in terms of exposure of hydrophobic surface was observed in the presence of Ca^2+^, a result confirmed by a subsequent independent study [14]. Nuclear magnetic resonance (NMR) spectroscopy (^1^H-^15^N-HSQC NMR) showed that in the absence of any cation (apo- condition), the structure of CIB2 is not ordered, while far-UV CD spectroscopy demonstrated residual α-helix secondary structure. The addition of either Mg^2+^ or Ca^2+^ induced a well-folded tertiary structure in CIB2, with many peaks overlapping despite the detectable structural difference among the cation-bound specific state [13]. An independent study was performed under similar conditions, after removing the poly-His-tag used in [13] for protein purification purposes, and led to very similar results, indicating that apo-CIB2 lacks tertiary structure and forms a molten globule state, as also confirmed by near-UV CD spectroscopy [14].

### 2.2. An Alleged Ca^2+^ Sensor? CIB2 Constitutively Binds Mg^2+^ but Has Low Affinity for Ca^2+^

In vitro studies showed that all CIB proteins respond to Mg^2+^ and Ca^2+^ stimuli in a relatively similar manner in terms of general structural features [13]; however, a recent investigation by our research group highlighted important peculiarities of CIB2 in cation-sensing [14]. NMR titration experiments reporting on the Ca^2+^ binding state of specific residues suggested that when starting from the apo-protein, EF3 is the first EF-hand in CIB2 to be occupied by Ca^2+^, followed by EF4. The variation of ^1^H-^15^N HSQC peak intensity for the EF4 residue Gly 162 upon Ca^2+^ titration experiments displays a sigmoidal shape, indicative of positive cooperativity [14]. However, in the presence of physiological levels of free Mg^2+^ (approximately 1 mM), EF3 seems never to be occupied by Ca^2+^ but rather constitutively Mg^2+^-bound. A Mg^2+^/Ca^2+^ exchange would thus possibly occur only in EF4 [14] (Figure 2).

While a similar conclusion was also previously drawn for CIB1 [22], there is an important difference between the two proteins. The apparent affinity for Ca^2+^ of CIB2 is as low as 500 μM [14] while significantly higher affinity was measured for Ca^2+^ binding to CIB1, (K_d_ = 0.5 μM for EF4; K_d_ = 1.9 μM for EF3 [22,23]), which makes its Ca^2+^ sensing capabilities similar to those of calmodulin [24]. Moreover, Mg^2+^ binding is limited to the EF3 loop in the case of CIB1 (K_d_ = 120 μM [22,23]), but it appears to involve both EF3 and EF4 in the case of CIB2 (apparent affinity: 290 μM [14]). The physiological consequence is clear when these values are considered in the context of intracellular levels of the two cations. Intracellular Ca^2+^ oscillates in the 0.1–10 μM range [25], and in specific cell compartments such as the outer segments of photoreceptors, where CIB2 has also been detected [2], it reaches even lower values [26,27]. On the other hand, free Mg^2+^ is generally constant in most cells and ranges in the 0.5–1 mM interval [28,29]. Overall, this suggests that CIB1 can indeed work as a Ca^2+^ sensor under physiological Ca^2+^ levels, while CIB2 would keep two Mg^2+^ ions constitutively bound to both functional EF-hands, being substantially insensitive to intracellular Ca^2+^ oscillations (Figure 2). It is to be noticed that based on a TNS fluorescence (2-p-toluidinylnaphthalene-6-sulphonate) assay, Blazejczyk et al. [4] reported a much higher affinity (K_d_ = 0.14 μM) for Ca^2+^ for GST-fused CIB2. This value is in contrast to data obtained with untagged CIB2 from direct spectroscopic analyses performed by NMR and CD titrations and indirect competition assays with 5,5′-Br_2_-BAPTA [14] and could be affected by artifacts induced by the interaction with the bulky (26 kDa) fused GST moiety.

The relatively low affinity for Ca^2+^ of CIB2, distinguishing it from CIB1, is not surprising if its primary structure is considered (Figure 1). The structure of CIB1 [19] reveals an optimal pentagonal bipyramid geometry offered by oxygen-coordinating Ca^2+^ in EF4 due to the presence of an Asn residue (N169) in the -X position of the coordinating loop and to a Glu residue in the -Z position (E172), which constitutes a bidentate ligand for Ca^2+^ that is broadly conserved among EF-hand motifs [30,31]. At odds, -X and -Z positions in CIB2 are occupied, respectively, by the side-chain-lacking G165 and by D168, which cannot act as a bidentate ligand due to the shorter side chain. This likely distorts the geometry of the whole loop and results in a lower affinity for Ca^2+^ [14]. Important differences are also found in EF3, which explains the lower affinity for Ca^2+^ of this loop for CIB2 compared to CIB1. A negatively charged residue (D118) occupies the Y coordinating position in CIB1, while a neutral one (N118) is substituted in CIB2, thus preventing a favorable coulombic interaction; moreover, the ninth position (-X) in the EF-hand loop is occupied by an Asn (N124) in CIB1 and by the least frequently observed Cys residue (C124), known for destabilizing some EF-hands [32]. The sequence of CIB2 is, therefore, not evolutionary optimized for Ca^2+^ binding.

The presence of physiological targets is known to significantly tune the metal affinity of Ca^2+^ sensors, often increasing it, such as in the case of myristoylated recoverin, a prototypical NCS protein in which the co-presence of the GRK1 target and the membrane is required to bring the Ca^2+^ sensitivity into the physiological range, thus enhancing over 100-fold the apparent Ca^2+^ affinity with respect to the isolated protein [33]. In the case of CIB2, the presence of a membrane-proximal peptide from the cytoplasmic domain of the α7B integrin was shown to double the apparent affinity for Ca^2+^ (K_d_^app^ = 200 μM vs. 500 μM in the absence of the target) [14]. While significant per se, this change does not support the physiological role of CIB2 as a Ca^2+^ sensor, unless the microenvironment of specific cell compartments and the co-presence of other factors such as myristoylation, membrane, or other supramolecular complexes will prove otherwise by enhancing the apparent affinity of CIB2 for Ca^2+^.

### 2.3. An Inter-Domain Allosteric Switch Regulates the Conformational Transitions of CIB2

Although with a lower affinity than CIB1, CIB2 binds Mg^2+^ and Ca^2+^ (the latter under non-physiological conditions), and this significantly stabilizes the structure of the protein. Thermal denaturation studies performed by monitoring the protein secondary structure content by far-UV CD spectroscopy indeed proved that apo-CIB2 is rather unstable, with a melting temperature (T_m_) of 35 °C, while the addition of Mg^2+^ or Ca^2+^ significantly increases the stability by enhancing the T_m_ of 11 and 8 °C, respectively [14]. In the co-presence of both cations, the T_m_ resembles that of Mg^2+^, confirming the minor stabilizing role of Ca^2+^ compared to Mg^2+^. The stabilizing effects of cations on the protein tertiary structure are even more apparent when analyzed by NMR, which also highlighted an important allosteric mechanism connecting EF3 with the non-functional EF1 motif. An inter-domain allosteric communication was detected between the EF3 binding loop and the residue E64, which is predicted to form an electrostatic interaction with R33, thereby contributing to the stability of the EF1 subdomain. This was clearly demonstrated by ^1^H-^15^N HSQC NMR spectra, where the chemical shift of N121 in EF3 and E64 in EF1 were found to show the same pattern upon Ca^2+^-titration experiments. The switch that allows CIB2 to acquire a functional conformation at physiological levels of Mg^2+^ seems, therefore, to be finely regulated by an allosteric, long-range communication connecting EF1 with EF3 (Figure 2a). Interestingly, E64 is mutated into an Asp (E64D) in a group of patients affected by Usher syndrome type 1 J [2]. Vallone et al. demonstrated that the apparently conservative E64D substitution breaks up such inter-domain communication, resulting in a protein that is unable to bind Mg^2+^, which is necessary to adopt the required physiological conformation, thus providing a first mechanistic explanation for the molecular basis of disease [14]. See Section 4.3 for further details.

### 2.4. CIB2 Is Monomeric, but It Could Dimerize in the Presence of a Target

Analytical size exclusion chromatography (SEC) is a valuable technique to study the hydrodynamic properties of proteins, and it can be used to assess the molecular weight (MW) of the eluting protein when a calibration curve is obtained in the same conditions for a number of globular proteins is available [34]. The apparent MW of CIB2 obtained by SEC in the presence of Mg^2+^ or co-presence of Mg^2+^ and Ca^2+^ was determined to be around 39 kDa, therefore significantly higher the theoretical MW of a monomer (22 kDa); this was initially interpreted as evidence of the dimeric nature of CIB2 [14]. A similar conclusion was supported by dynamic light scattering (DLS) measurements. A subsequent study employing a variety of mass spectrometry (MS)-based techniques, including native ESI-MS, MALDI-TOF-MS, and cross-linking/MS integrated with novel SEC and DLS experiments based on a more accurate selection of the heterogeneous components of the elution bands, demonstrated that CIB2 was monomeric under all tested conditions [12]. A comparison with results obtained with recoverin and calmodulin suggested that the apparent MW, extrapolated by analytical SEC for a Ca^2+^ sensor protein and based on the hydrodynamic radius, can significantly differ from the real value when the protein has a high hydrophobic solvent-accessible surface, such as in the case of recoverin and CIB2 [12]. Analytical SEC could then be driven to erroneous conclusions as to the oligomeric state of the protein [14]. This can explain the apparent contradiction with previous studies that detected CIB2 dimers based on FRET and co-immunoprecipitation. In these studies, CIB2 was fused to GFP [2,35] or tdTomato fluorescent protein [2]; its uncommon hydrodynamic properties, together with a possible interference of the bulky fusion constructs, may have led to the erroneous detection of a dimer [12]. The most recent and thorough dedicated investigation performed with untagged purified CIB2 seems to exclude the existence of CIB2 dimers under conditions mimicking the physiological ones [12]. In this respect, the oligomeric state of CIB2 resembles that of CIB1. Ca^2+^-bound CIB1 was indeed found to be monomeric in the crystallographic structure reported by Gentry et al. [19], and this finding has been supported by NMR diffusion, SEC, and sedimentation equilibrium experiments [36,37]. The head-to-tail dimer reported by another crystallographic study could result from the specific conditions for crystal formation, which included a GSH moiety at the N-domain [38].

Surface plasmon resonance (SPR) spectroscopy also excluded the dimerization of CIB2 over a broad range of conditions, including selective incubation with Mg^2+^ and Ca^2+^ [12], but suggested an interesting mechanism of binding to integrin. The formation of a protein–peptide complex between CIB2 and a peptide from the α7B integrin was shown to possibly drive the binding of a second CIB2 molecule; the process seems to be kinetically favored in the sole presence of Mg^2+^ [12]. Although this hypothesis awaits confirmation, the mechanism of target-induced CIB2 dimerization would explain the 2:1 protein:peptide stoichiometry detected for the same interaction in a previous fluorescence study [14], and it is tempting to speculate that it might play a role in the integrin signaling mediated by CIB proteins.

### 2.5. CIB2 Myristoylation

Myristoylation is a post-translational modification operated by the N-myristoyl transferase that, in vivo, covalently binds a 14-carbon saturated fatty acid (myristoyl moiety) to the N-terminal glycine of proteins harboring the consensus sequence MGXXXS/T [39]. CIB1, CIB2, and CIB3 have the optimal consensus sequence (Figure 1) and are likely to be myristoylated in vivo, while CIB4 is not. The binding of Ca^2+^ ions to some NCS proteins triggers the so-called “myristoyl switch” mechanism, which extrudes the myristoyl moiety from a hydrophobic cleft in the protein milieu to a fully solvent-exposed state, thus allowing membrane-binding and permitting specific cell localization and target interaction [40].

It is not clear whether CIB1 undergoes a Ca^2+^-induced myristoyl-switch, as contrasting conclusions have been reported by different groups. Jarman et al. [41] concluded that the myristoyl switch mechanism occurs in CIB1, and it is necessary to translocate the sphingosine kinase 1 (SK1) target to the membrane; in contrast, according to Blazejczyk et al., the myristoyl moiety of CIB1 is likely solvent-exposed rather than buried within the protein, regardless of its Ca^2+^ binding state [42]. Whether or not the Ca^2+^-induced switching mechanism occurs, the myristoylation of CIB1 is still important for protein stability and membrane targeting [43,44], as well as shuttling its binding partners to the membrane [41,45]. Myristoylation-deficient CIB1 variants retain a high affinity for target proteins and peptides both in vitro and in vivo [42,44,46,47], thus suggesting that the myristoyl group is not specifically involved in target recognition.

Much less information is available about the myristoylation of CIB2. Blazejczyk et al. [4] expressed GFP-tagged CIB2 and a myristoylation-blocking variant (^G2A^CIB2-GFP) in COS-7 cells in the presence of radioactive ^3^H-myristic acid and compared the effect of CIB2 myristoylation with that of VILIP1-GFP [48], known to undergo the Ca^2+^-induced myristoyl switch, thereby resulting in specialized membrane compartment localization. In contrast to what was observed for VILIP1-GFP, no Ca^2+^-dependent translocation was detected for CIB2-GFP, which was interpreted as a lack of Ca^2+^-induced myristoyl switch [4]. Moreover, localization of CIB2 was essentially limited to the crude membrane fraction, regardless of the tag (CIB2-FLAG gave the same results) and, surprisingly, of myristoylation as the same behavior was detected for ^G2A^CIB2-GFP [4]. This would exclude a direct association of CIB2 to the membrane by myristoyl anchoring and rather suggests that the binding is mediated directly by lipids or by an interaction with other proteins. Within COS-7 cells, CIB2 co-localizes with the Golgi apparatus and not with the nucleus, and the absence of myristoylation does not affect this pattern [4].

An independent study by Zhu et al. [49] confirmed that CIB2 does not undergo a Ca^2+^-induced myristoyl switch and blocks the agonist-induced membrane translocation of SK1, at odds with the activity of CIB1. Experiments were performed with HA-tagged CIB2 in HEK293 cells labeled with ^3^H-myristic acid and showed that the interaction of CIB2 with SK1 was independent of the presence of Ca^2+^ or Mg^2+^. Further experiments are needed to elucidate the biological role of CIB2 myristoylation, and biophysical studies may clarify the structural and mechanistic aspects that remain somewhat unclear.

## 3. Broad Expression Pattern and Interaction Promiscuity: Functional Plasticity of CIB2

Recent findings support the noteworthy functional plasticity of CIB2 and a variety of interactors (Figure 3). We summarize in the next paragraphs the salient molecular features of such interactions and their biological implications. A separated section is dedicated to the role of CIB2 in hearing and related diseases.

### 3.1. Interaction between CIB2 and Integrins

As evoked by the name, CIB2 interacts with heterodimeric membrane proteins called integrins, responsible for the inside-out and outside-in signaling in cells, thus playing key roles in development, immune responses, and hemostasis. Many studies were performed to characterize CIB2–integrin interactions in terms of their specificity and kinetics, structure–function properties, and biological relevance. So far, two integrins have been found to interact with CIB2: αIIbβ3 [8] is expressed by platelets and megakaryocytes and, apparently, a common target for all CIB family members, at odds with α7Bβ1D [3], which seems to be CIB2-specific and is expressed in skeletal muscles.

Based on the evidence that CIB1 works as a negative regulator of agonist-induced αIIbβ3 activation in murine megakaryocytes [50] and that binding to such integrin is necessary for the proper spreading on fibrinogen [51], DeNofrio et al. [8] generated a CIB1 knockout mouse (*CIB1^−^*^/^*^−^*), expecting dramatic effects on platelets function. Surprisingly, they found that *CIB1^−^*^/^*^−^* and *CIB1*^+/+^ mice were very similar to one another in terms of platelets aggregation, outside-in signaling, and cell morphology, suggesting that a compensatory effect might be played by other CIB family members. This was confirmed via immunoassays, where a peptide encompassing the C-terminal portion of αIIb integrin (see Table 1 for the sequence) was shown to interact with both recombinant CIB2 and CIB3, thus explaining, at least partly, the lack of deleterious consequences in the *CIB1^−^*^/^*^−^* model.

At the same time, an independent research group detected a specific interaction of CIB2 with the α7Bβ1D integrin [3]. By immunofluorescence experiments, Hager et al. identified similar expression patterns of CIB2 and α7B integrin in the skeletal muscle of WT mice [3]. Extending the analysis to a mouse model lacking the expression of laminin α2 chain (dy^3K^/dy^3K^), causative of the rare autosomal recessive disease congenital muscular dystrophy type 1A (MDC1A), they found a 2.6-fold reduction of α7B and CIB2 expression that could be partially restored by inducing the laminin α1 chain expression [3]. In vitro binding assays confirmed the specific interaction of CIB2 with both α7B and β1D integrin subunits, with a stronger binding to α7B. Intrinsic tryptophan fluorescence (ITF) titrations, using the peptide encompassing the cytosolic portion of α7B (here indicated by α7B_C) permitted the first estimation of the equilibrium dissociation constant, resulting in relatively high affinity (K_d_ = 304 nM; see Table 1).

The diversity of integrins found to interact with CIB2 raises the question of whether a common region in the integrin structure can trigger the interaction. In this respect, Huang et al. [13] first confirmed the CIB2-αIIb interaction, which appears not to be modulated by cations, and found that, in vitro, the interaction with the cytosolic portion appeared weaker than that with the membrane-proximal region α7B_M (see Table 1), which was slightly reduced in the presence of Ca^2+^.

The experiments described above were performed using either mouse or human CIB2 (with or without a His-tag) and integrins. Other experimental differences include temperature (20 °C [3] vs. 25 °C [13]) and working buffers, which, in some cases, had non-physiologically-low ionic strength [13]. This is a crucial factor that may affect the physico-chemical properties of CIB2, given its high (60% of the total) hydrophobic solvent-accessible surface [12] that may be stabilized by the presence of salt. In an attempt to provide robust comparisons under experimental conditions mimicking the physiological ones, Dal Cortivo et al. [12] and Vallone et al. [14] characterized the CIB2-α7B interaction using recombinant human proteins and focusing on the specificity, affinity, kinetic and structural features. SPR spectroscopy confirmed the previous observation by Huang et al. as to the stronger binding to α7B_M than α7B_C [13]. Moreover, titrations performed with increasing concentrations of the α7B_M peptide showed that Ca^2+^ decreased the apparent affinity for CIB2 compared to the sole presence of Mg^2+^. As discussed in Section 2.4, based on the SPR results, Dal Cortivo et al. proposed an interesting mechanism of α7B_M-induced CIB2 dimerization [12], which is in line with the previous observation by Vallone et al. [14] of a 2:1 CIB2:peptide stoichiometry detected in fluorescence titrations.

### 3.2. CIB2 and Ovarian Cancer

Ovarian cancer (OC) is the most lethal gynecological malignancy, characterized by resistance to chemotherapy treatments [52] and the absence of specific molecular markers for early diagnosis [53]. Several studies have described CIB1 as a pro-oncogenic protein: the myristoyl-switch mechanism is responsible for the Ca^2+^-dependent translocation of SK1 from cytosol to the plasma membrane, where it phosphorylates sphingosine to sphingosine-1-phosphate [41], an important oncogenic messenger that promotes the metastatic behavior. Interestingly, the exhaustive study by Zhu et al. found that CIB2 acts as a negative regulator of the oncogenic signaling in OC [49]. CIB2 competes with CIB1 for SK1 binding independently on the N-terminal myristoylation. This drove the authors to conclude that, as opposed to CIB1, CIB2 is an endogenous inhibitor of the pro-oncogenic pathway mediated by SK1. Moreover, the analysis of the expression levels of *CIB2* in different human tumors highlighted its downregulation in OC, flanked by the higher expression of SK1. The tumor-suppressor role of CIB2 was investigated using specific OC cell lines, in which the re-expression of CIB2 led to hindered neoplastic growth, a 50% reduction in cell migration, and a higher response to carboplatin treatment. Interestingly, the downregulation of CIB2 was found to be unrelated to tumor stage or grade, making this small protein a good candidate for early-stage diagnosis and a potential target to improve the efficacy of chemotherapy.

### 3.3. Involvement of CIB2 in HIV Viral Infection

The infection–proliferation cycle of human immunodeficiency virus-1 (HIV-1) is constituted by different steps, including (i) membrane attachment (cell-free or cell-to-cell transmission); (ii) receptor (CD4 or CD5) and co-receptor (CXCR4 in T-cells and CCR5 in macrophages) binding; (iii) membrane fusion (triggered by proteins such as CXCR4, CCR5, CD3, LFA-1, α4β7, tallin); (iv) transport of viral capsid into the nucleus; (v) nuclear transport, integration, and transcription; (vi) viral protein synthesis; and (vii) novel viral particles budding [54]. The identification of host proteins fundamental for viral replication by small interfering RNA (siRNA) and short hairpin RNA (shRNA) screening represents a good strategy for designing new treatments against the infection. In this regard, two independent studies [55,56] have identified CIB2 as a good candidate. Godinho-Santos et al. [54] investigated the potential role of CIB1 and CIB2 in HIV-1 cell entry. This well-designed and exhaustive study is based on knockingdown *CIB2* (and *CIB1*) in Jurkat cells, HeLa cells, CD4^+^ T-cells (from human peripheral blood), and PM1 cells by RNA interference technique. The specific silencing procedure did not alter cell viability, thus suggesting that a normal concentration of CIB2 was not required for cell survival. In a second step, cells were transduced with HIV-1, and each infection step was singularly analyzed. Interestingly, it was found that CIB2 suppression affects the early stages of HIV-1 entry, such as receptor-mediated viral binding and membrane fusion, both in cell-free and cell-mediated viral interaction [54]. A lower expression of membrane proteins such as CXCR4, integrin α4β7, and CCR5 (in PM1 cell line) was indeed found in CIB2-knocked down T-cells, suggesting a potential role of at least one of these proteins in promoting viral infection. Similar results were obtained with CIB1, suggesting that the two paralog proteins act as host helper factors in HIV-1 entry and replication. The finding that normal concentrations of CIB2 facilitate the virus entry could be used to develop treatments in the early stages of the viral infection.

### 3.4. CIB2, mTORC1 Signaling, and Autophagy

In a very recent work, Sethna et al. [15] reported a novel role for CIB2, which has deep implications for cell signaling and, in particular, for the mechanisms underlying the development of age-related macular degeneration (AMD), a multifactorial disorder of the macula, causing central vision loss, affecting 25% of people over 75 years old [57]. Using three different mouse lines with both complete loss (homozygous) and haploinsufficiency (heterozygous) of *CIB2* in the retinal pigmented epithelium (RPE), Sethna et al. proved that abnormally low expression levels of CIB2 induced in mice age-related features similar to those found in dry AMD patients, which resulted in attenuated electroretinogram (ERG) amplitudes, indicative of visual loss. This finding is at odds with results previously reported by Michel et al. [58] in *CIB2*^−/−^ mice that did not report vision deficits. Detection of fewer lysosomes and autophagy proteins were accompanied by defects in photoreceptor outer segment clearance and sub-RPE lipid deposits, similar to the drusen found in humans. Increased lipid droplets and esterified cholesterol were associated with both the global and RPE-specific lack of CIB2 and could be partly rescued by exogenous retinoids [15].

Interestingly, a comparison between mouse models and RPE cells cultured from AMD patients revealed that in both systems, the reduced levels of CIB2 correlated with hyperactivity of mTORC1, a fundamental negative regulator of autophagy, suggesting that CIB2 negatively regulates mTORC1 signaling [15]. mTORC is known to be activated by the GTP-bound form of the small GTPase Rheb [59]. Sethna et al. found that CIB2 preferentially binds to the nucleotide-free (inactive) form of Rheb, although the interaction was also detected for GDP-Rheb and GTP-Rheb; the authors suggested that CIB2 functions as a “co-factor” that maintains Rheb in an inactive state. The study, however, does not provide biochemical data, such as nucleotide exchange assays, which would be essential to fully support this conclusion; it only provided indirect evidence. Current biochemical data do not support the role of CIB2 as a guanine nucleotide exchange factor (GEF) or guanine dissociation inhibitor for Rheb; therefore, further dedicated experiments are needed to better clarify the molecular scenario.

### 3.5. Putative Roles of CIB2 Investigated by OMICS Techniques

Omics techniques represent powerful tools that allow the discovery of new alleles, expression profiles, or DNA modifications that may be used to gain knowledge on disease-associated variants and their incidence in the population. In this regard, three recent works have presented relevant results on *CIB2*. The first account by Peasad et al. [60], designed to find variants potentially causative of autism spectrum disorder (ASD), reported a deep analysis of copy number variations (CNVs) on a cohort of 696 unrelated autistic Canadian patients. One of the ASD-specific CNVs was a maternally inherited duplication at 15q25.1, found in three unrelated cases (two men and one woman), that led to the disruption of one *CIB2* exon.

A similar approach was used by Mangino et al. [61], who combined genomics, transcriptomics, and epigenomics to identify in a cohort of 1505 healthy twins of European descent a single nucleotide polymorphism in *CIB2*, associated with central arterial stiffness. Reduced methylation of *CIB2* may lead to increased protein expression, which could affect the process of vascular calcification through a mechanism yet to be clarified [61].

Finally, transcriptomic analysis was performed by Vastagh et al. [62] on female mice during proestrus, the period prior to the estrous phase, associated with a rise of circulating estradiol [63]. The study aimed to identify differentially expressed genes from gonadotropic-releasing hormone (GnRH) neurons. The authors found 43 modulated genes, 37 expressing for K^+^, Na^+^, and Cl^−^ channels, and 8 related to calcium-homeostasis, among which *CIB2* was found to be downregulated.

In general, care should be taken in defining a mutation as associated or causative of a specific phenotype. Mechanistic details on the putative involvement of CIB2 in the aforementioned processes have not been analyzed so far, but these results represent an interesting starting point for future dedicated studies.

## 4. Involvement of CIB2 in Hearing and Deafness

Following the identification of *CIB2* as a causative gene in Usher syndrome type 1 J and non-syndromic deafness [2], a number of investigations have established a prominent role of CIB2 in hearing physiology and disease [9]. Interestingly, although integrins are important elements for the regulation of anatomical and biochemical processes of stereocilia, such as maturation, differentiation [64], and cytoskeleton structure assembly [65], so far, no direct evidence exists of CIB2–integrin interaction in the hearing system. The role of CIB2 in this highly specialized neuronal sensory transduction system could therefore be different from that in muscles or platelets, as summarized in the following paragraphs.

### 4.1. CIB2 as a Member of the Mechanoelectrical Transduction Complex

Mechanoelectrical transduction (MET) is the process occurring in hair cells responsible for the conversion of sound waves into electrical signals. The cascade is stimulated by the physical stretching of the tip-links that mechanically open the MET channels, leading to fluctuations in the concentration of free Ca^2+^, which eventually depolarize the cells and propagate the signal to downstream neurons.

Many studies have focused on the analysis of MET channels in different species; for comprehensive reviews, see [11,66]. It is well known that MET channels are complex systems formed by many proteins, of which transmembrane channel-like 1 and 2 (TMC1 and TMC2, respectively) are the major pore-forming units in mammals [67]. However, information on the presence of auxiliary proteins and on the way they work or modulate the MET function is still sparse.

Giese et al. [35] and Wang et al. [68] used transgenic mice to investigate the functional role of CIB2 in hair cells and its potential relation with MET channels. By replacing the wild-type protein with truncated forms of CIB2 encompassing residues 1–66 (*CIB2**^tm1a^*) and 1–109 (*CIB2*^−/−^), they independently found that CIB2 expression was required for the proper functioning and maintenance of the mature stereocilia as the absence of protein expression deeply perturbs the staircase organization of the hair bundle, which is crucial for the generation of the MET currents. This led to profoundly deaf mice characterized by compromised auditory evoked brainstem responses (ABR) and distortion product otoacoustic emission (DPOAE). Giese et al. [35] explained this evidence with the fact that MET channels were properly localized but constitutively closed, thus excluding any role of CIB2 in the proper channel localization on the tips of stereocilia. This conclusion is, however, under debate since a later investigation by Liang et al. [20], using a mouse model co-expressing truncated CIB2 (*CIB2**^tm1a^*) and TMC1-HA or TMC2-MYC, demonstrated that MET channels were retained in the cell bodies and only the re-expression of CIB2 restored their proper localization. These lines of evidence led the authors to conclude that TMC1/2 and CIB2 interact directly.

To identify the specific region of such interaction, Giese et al. generated truncated forms of TMC1 encompassing the N-terminal 1–193 region [35]. With a similar goal, Liang et al. scanned the full-length channel [20]. Combining FRET measurements on transfected cells and co-immunoprecipitation assays, two possible regions of interaction were identified. The 81–130 TMC1 region was identified in both studies, although Liang et al. found the interaction with CIB2 to be relatively weak for this region, and the recognition was found not to be modulated by Mg^2+^ and Ca^2+^ [20]. In addition, the latter authors found another TMC1 portion, encompassing residues 305–344, which binds CIB2 with higher affinity, probably via hydrophobic interactions. Despite the differences in the resulting interacting regions, these studies have demonstrated that CIB2 is fundamental for the proper function of the MET channel. The functional interpretation of the results was significantly different in the two reports. Giese et al. [35] proposed that in the absence of functional CIB2, the MET channels would be properly located but not functional. Conversely, Liang et al. [20] found that the re-expression of CIB2 via injectoporation is necessary to restore the MET currents and, therefore, the native localization of the channel at the tip of the stereocilia. This latter aspect was deeply investigated by the authors, who found that CIB2 regulated the TMC1 opening probability (P_0_) at rest in a Ca^2+^-dependent fashion.

In conclusion, CIB2 may work as a channel regulator, either as a tension-release element [35] or as an intrinsic component of the channel, and, therefore, is pivotal for its function [20]. The actual scenario remains unclear when the conclusions of the two investigations are considered. Findings by Liang et al. [20] are consistent with CIB2 not participating in the adaptation process, while according to Giese et al. [35], the loss of Ca^2+^-buffer properties due to the lack of fully functional CIB2 could result in the overgrowth of transducing stereocilia. This is an intriguing hypothesis if the binding of CIB2 to a target is shown to bring the Ca^2+^ sensitivity of CIB2 into the physiological range, a condition that seems to be excluded for the sole protein based on in vitro investigations [14].

### 4.2. CIB2 Interacts with BAIAP2L2

The MET process involves a complex molecular scenario, where many proteins and protein complexes work in concert to achieve proper signal transduction. Among these, “row protein complexes” are groups of proteins localized on the tip of stereocilia, whose composition differs based on the stereocilia length. Rows of stereocilia, each of ascending height, with Row 1 being the tallest, are arranged to produce a bundle with a precise staircase-like morphology.

It is known that a deficiency in Row1 proteins results in abnormally short stereocilia [69,70,71,72]. Much less is known about Row2–3 protein complexes, localized in the shorter stereocilia. Yan et al. [73] published a well-designed study focused on BAIAP2L2 (BAI1-associated protein 2-like 2; also known as Pinkbar), a BAR-domain containing protein whose function is still unknown but has been recently identified as a non-syndromic hearing loss gene [74]. The study demonstrates that BAIAP2L2 is a component of the Row2–3 protein complex, which co-localized with the MET channels on the tips of stereocilia. This finding, together with the MET current impairment found in *BAIAP2L2*^−/−^ mice, prompted the authors to investigate the potential interaction between BAIAP2L2 and the soluble component of MET channels CIB2. Co-immunoprecipitation and colocalization experiments on transfected cells demonstrated the specific interaction between CIB2 and BAIAP2L2, and whole-mount immunostaining on *CIB2*^−/−^ knockout mice highlighted that the lack of CIB2 prevents the proper localization of BAIAP2L2 [73].

Taken together, all these lines of evidence further support the plasticity of CIB2 in terms of structure and function as well as its fundamental role played in hearing and, thus, in deafness.

### 4.3. Mutations in CIB2 Are Associated with Deafness

At present, 13 identified mutations in the gene encoding CIB2 have been associated with different forms of hearing impairment, including five nonsense mutations with an ultra-rare frameshift [75] and eight missense mutations (Figure 1 and Table 2). Since the loss of CIB2 is deleterious for the proper MET process, it is expected that nonsense mutations that generate truncated forms of CIB2 may severely affect hearing. Four out of five of the known nonsense mutations are homozygously inherited and have been found in families of the following ethnicity: Turkish [75], Palestinian Arabian [58] (p.Tyr110*); Arabic [76] (p.Arg33*); Tunisian [77] (p.Arg104*); and Iranian (p.Gln12* [58]). The other two variants are heterozygous compounds found in Dutch [78] (p.Arg66Trp + Arg33*) and European [75] (p.Arg66Trp and p.Glu100*) families. A common feature among the different cohorts is the profound pre-lingual bilateral hearing loss, without vestibular impairment or retina abnormalities, which suggests that mutations in *CIB2* are causative of autosomal recessive non-syndromic hearing loss (ARNSHL or DFNB48).

Functional effects exerted by missense point mutations are difficult to predict since amino acid substitutions may perturb protein stability, the affinity for the cations, and allosteric effects or modify the interaction with targets. In this respect, a good example is the conservative point mutation E64D, found in consanguineous Pakistani families by Riazuddin et al. [2] and defined as causative of Usher syndrome 1J (see below). Data available in the literature about this variant are discordant. Riazuddin et al. [2] used ratiometric Ca^2+^ imaging to infer the affinity for Ca^2+^ of the variant and concluded that E64D CIB2 shows WT-like properties, thus suggesting unaltered Ca^2+^-affinity. In contrast, Vallone et al. [14] demonstrated that Ca^2+^ and Mg^2+^ affinity were deeply compromised for E64D CIB2, where the mutation blocks the protein in a molten globule state that makes it impossible to reach the proper structure required for its function at physiological levels of cations (Figure 2b). The substantially different behavior detected in the two independent studies may be due to the different experimental design: while Riazuddin et al. [2] used a more complex system consisting of transiently transfected cells (COS-7), Vallone et al. [14] used the recombinant purified protein and performed in vitro spectroscopy. Due to the high plasticity of CIB2, it is possible that in eukaryotic cells, CIB2 interacts with other proteins, which may modulate its affinity for cations and its intracellular dynamics.

Five other point mutations are causative of DFNB48: F91S, the most common substitution in cases of sensorineural hearing loss [2,35,78,79], R66W [78], C99W [2,35,79], I123T [2,20], and R186W [20,80,81]. Biochemical studies for these variants are less exhaustive compared to the E64D variant. Using ratiometric Ca^2+^ imaging, Riazuddin et al. [2] and Patel et al. [81] found that: (i) CIB2 is an inhibitor of Ca^2+^ release from the IP3-receptor (depletion of 40% of the total Ca^2+^ with respect to control), (ii) F91S, C99W, and R186W induced a higher Ca^2+^ release, and (iii) I123T induced a lower release. Based on these findings, the authors suggested compromised and enhanced Ca^2+^ affinity, respectively, for the pathogenetic variants. It should be noted, however, that the differential modulation of the IP3-receptor by CIB2 variants per se does not necessarily imply a change in affinity for Ca^2+^ of the single protein but rather a change in sensitivity of the whole complex. This has been clearly demonstrated for E64D CIB2, which is essentially capable of binding α7B_M with a wild–type–like affinity while being incapable of binding Mg^2+^ and/or Ca^2+^ [14].

**Table 2 ijms-23-03552-t002:** Summary of the known mutations in the CIB2 encoding gene.

cDNA	Protein	Type	Genotype	Ethnicity	Phenotype	Refs.
c.34C > T	p.(Gln12*)	Nonsense	Homozygous	Iranian	NSHI	[58]
c.97C > T	p.(Arg33*)	Nonsense	Homozygous	Arabic	HSHI	[76]
c.196C > T + c.300_309	p.(Arg66Trp) + p.(Glu100*)	Missense + null allele	Heterozygous compound	European	HSHI	[75]
c.196C > T	p.(Arg66Trp)	Missense	Homozygous	Dutch	NSHI	[78]
c.192G > C	p.(Glu64Asp)	Missense	Homozygous	Pakistani	USH	[2]
c.97C > T + c.196C > T	p.(Arg33*) + p.(Arg66Trp)	Null allele + missense	Heterozygous compound	Dutch	NSHI	[78]
c.223G > A	p.(Val75Met)	Missense	Homozygous	European	NSHI	[75]
c.272T > C	p.(Phe91Ser)	Missense	Homozygous	Pakistani	HSHI	[2]
c.297C > G	p.(Cys99Trp)	Missense	Homozygous	Pakistani	HSHI	[2]
c.310C > T	p.(Arg104*)	Nonsense	Homozygous	Tunisian	NSHI	[77]
c.311G > A	p.(Arg104Gln)	Missense	Heterozygous	Not reported	PP	[82]
c.330T > A	p.(Tyr110*)	Nonsense	Homozygous	Turkish	HSHI	[58,75]
c.344A > G	p.(Tyr115Cys)	Missense	Homozygous	Iranian	HSHI	[75]
c.368T > C	p.(Ile123Thr)	Missense	Homozygous	Pakistani	HSHI	[2]
c.556C > T	p.(Arg186Trp)	Missense	Homozygous	Hispanic Caribbean Nigeria	NSHL PP	[80,81]

Abbreviations: NSHI, non-syndromic hearing impairment; USH, Usher syndrome; PP, predicted pathogenic. * Asterisks refer to non-sense mutation, where a stop codon is inserted at the indicated position.

Concerning the effects of mutations on the cellular localization, ex vivo and immunofluorescence experiments using native or GFP-tagged CIB2 (F91S-GFP, R66W-GFP, WT-GFP), both in WT and knockout animals, demonstrated that WT CIB2, F91S, and R66W were properly localized on the tips of stereocilia [75], which was confirmed both in OHC (outer hair cells), IHC (inner hair cells) and vestibular explants [78]. As to the capability of target recognition, Giese et al. [35] found that E64D, F91S, and C99W CIB2 variants were not capable of binding full-length TMC1 in co-transfected cells. Conversely, Liang et al. [20] demonstrated that in mouse models expressing tagged full-length TMC proteins (TMC1-HA and TMC2-MYC), the presence of E64D, I123T, and R186W mutations affects the proper localization of the channels on the tip of the stereocilia. Furthermore, they proved that R186W CIB2 does not permit the regulation of P_0_ of TMC1 channels in a Ca^2+^-dependent fashion.

A novel point mutation in *CIB2* was recently found in two Iranian families by Booth et al. [75], consisting of the replacement of Tyr115 with Cys (Y115C), although no structural, functional, or biochemical information is available yet. Finally, the V75M mutation, firstly identified in 2015 by Booth et al. [83], was found not to segregate with deafness, but, if inherited together with a missense point mutation (G228R) in PDZD7, it may be responsible for a more severe phenotype. Later, Booth et al. [75] confirmed the presence of this mutation in a European family, where two consanguineous parents generated siblings affected by ARNSHL. Not much is known about the pathogenic mechanisms, except for the hypothesis that the amino acid replacement may affect the interaction with TMC1/2 [75]. Finally, Fuster-Garcia et al. [82] published a study reporting 42 putative pathogenic mutations in genes associated with syndromic and non-syndromic deafness, and, among them, R104Q was found in *CIB2*, and the mutation was predicted to be possibly pathogenetic.

Treatment of sensorineural deafness is normally limited to cochlear implants, which does not always result in satisfactory outcomes. In this regard, Nishio et al. [84] investigated whether the genetic basis of deafness could help in the development of specific interventions and/or to predict the effectiveness of cochlear implants. Using RNA-sequencing following RNA extraction from different portions of the inner ear of C57BL mice, they found the expression level of *CIB2* to be 92-fold higher compared to normalizer genes in spiral ganglion cells, responsible for the communication between the cochlea and the central nervous system. Based on this evidence, the authors hypothesized that mutations in genes preferentially expressed in spiral ganglion cells may be associated with their degeneration and affect the poor outcome of the cochlear implant. This is an intriguing hypothesis, which needs further investigation to assess its real impact on personalized treatments of sensorineural deafness.

### 4.4. CIB2 and Usher Interacome

Usher syndrome (USH) is a rare autosomal recessive disease characterized by deafness with early-onset vestibular dysfunction and blindness due to retinitis pigmentosa (RP) at late stages [85]. The involvement of the two senses responsible for the income of most of the information from the environment makes this disease extremely disabling. USH is classified into three types based on the severity and age of onset. The group of genes/proteins that have been associated with Usher syndrome forms the so-called Usher interactome; for a comprehensive review, see [86,87].

The association between CIB2 mutations and USH has been debated. In 2012, Riazuddin et al. [2] proposed CIB2 as a member of the Usher interactome based on the outcome of co-immunoprecipitation assays that suggested the capability of CIB2 to interact with whirlin and myosin VIIa. This, together with the identification of E64D CIB2 in a Pakistani family affected by deaf–blindness, led the authors to conclude that this variant is causative of USH type 1J (USH1J). Moreover, the authors found that flies (*Drosophila melanogaster*) lacking CIB2 showed decreased photoresponses and progressive photoreceptor degeneration. However, three studies published between 2017–2018 seem to contradict this finding. Booth et al. [75] analyzed five families carrying mutations in *CIB2* and affected by pre-lingual, bilateral severe-to-profound hearing loss from different origins (Iranian, Turkish, and European), finding no evidence of RP or vestibular impairment. Michel et al. [58] then demonstrated that CIB2 is fundamental for the function and survival of hair cells in mature stages. Its absence affects the localization of proteins belonging to the Usher interactome, such as whirlin, integrins, and CIB1, but at the same time, its role in the retina appears to be negligible. To confirm this finding, the authors reported sequencing data on two families from Iran and Pakistan, characterized by the presence of CIB2 as null alleles (Y110* and Q12*). While affected by profound hearing loss, patients showed no vestibular or retinal impairment, thus confirming that the absence of CIB2 was not a direct cause of Usher syndrome. Finally, in 2018, Journet et al. [88] performed a meta-analysis of NGS data from 11 independent studies and found that mutations in *CIB2* were never detected in Usher-syndrome-declared cases. Taken together, all these lines of evidence call into question whether CIB2 and its point mutations belong to the Usher interactome.

## 5. Conclusions

The broad expression of CIB2 and the many interactors found in various tissues may reflect either a universal biological role of the protein yet to be uncovered or, rather, unusual functional plasticity. Although further biochemical and biophysical studies are needed to clarify if and under what conditions CIB2 could act as a Ca^2+^ sensor, some lines of evidence suggest that its functional plasticity is significantly broad, perhaps comparable to that of the ubiquitous signal transducer calmodulin. Calmodulin regulates more than 300 intracellular targets, each interaction being characterized by specific binding sites, Ca^2+^-dependency, and target affinity [89,90]. Whether the ancient evolutionary relationship between the two cation-binding proteins reflects similarly broad roles is an intriguing question that should drive future research.

## Figures and Tables

**Figure 1 ijms-23-03552-f001:**
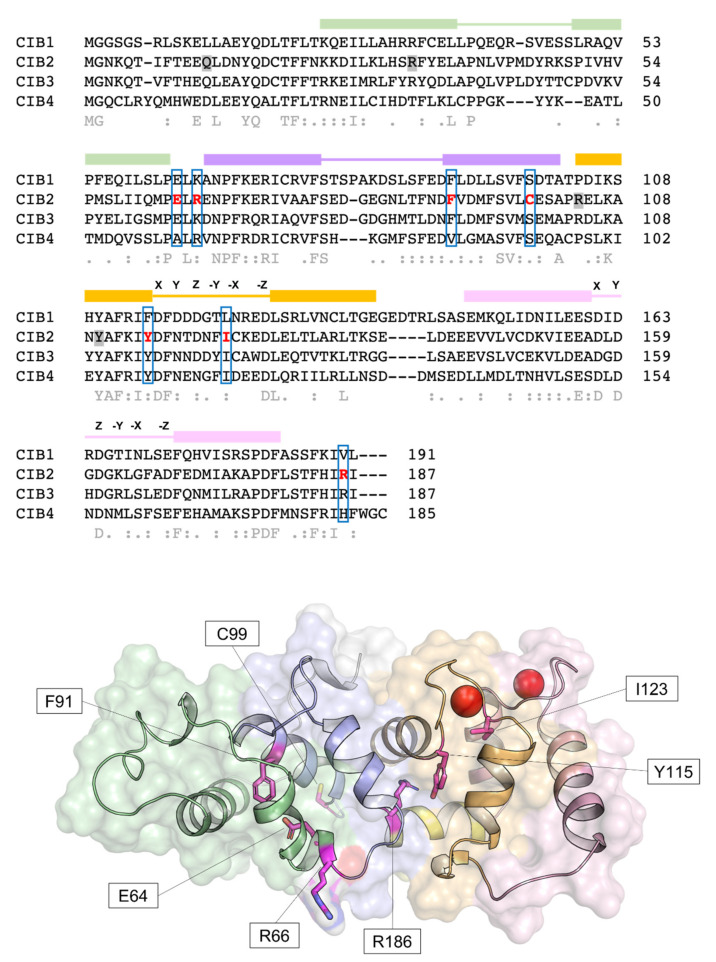
Sequence and structural properties of CIB2 and its pathogenetic variants. Top: multiple amino acid sequence alignment of human CIB1, CIB2, CIB3, and CIB4 performed by T–Coffee (https://tcoffee.crg.eu/apps/tcoffee/index.html, accessed on 22 February 2022). Consensus sequence and similarity descriptors are reported in grey text. Residues target of pathogenetic point mutations are grey-shaded in case of nonsense mutations or marked in red and blue-boxed in case of missense mutations (see Section 4.3). EF-hand motifs are displayed by colored boxes, and residues involved in Ca^2+^-coordination are indicated by letters referring to the canonical pentagonal bipyramidal geometry on the respective loops. Bottom: Representation of the three-dimensional structure of human CIB2 based on the homology model used in [14]. Cartoon and surface representation are superposed. The N-terminal region is colored gray, while the C-terminal region (helix 10) is colored yellow. EF1, EF2, EF3, and EF4 are colored green, slate blue, orange, and pink, respectively. Residue targets of pathogenic missense mutations are represented by magenta sticks, and Ca^2+^ ions are represented by red spheres.

**Figure 2 ijms-23-03552-f002:**
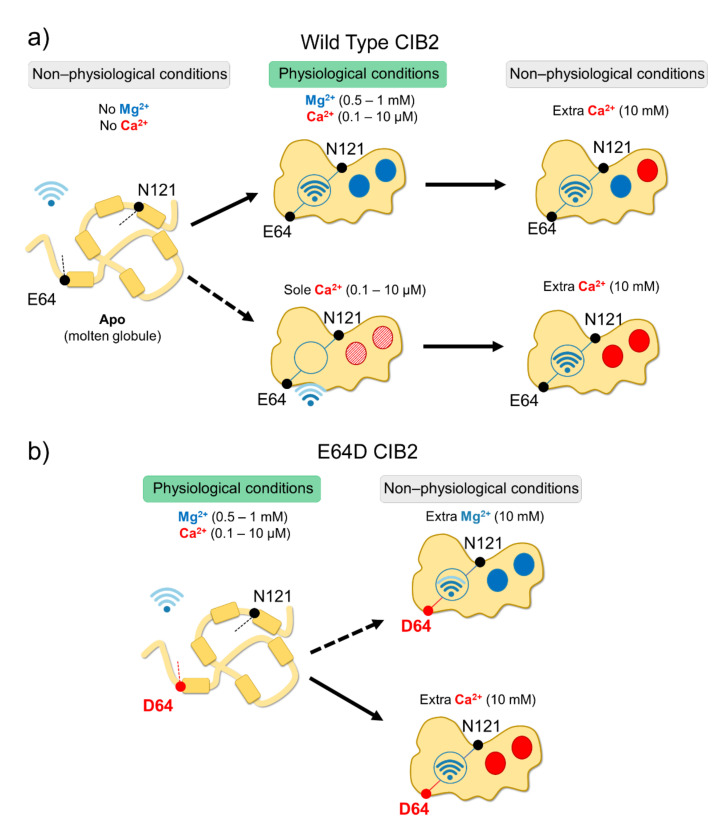
Schematic representation of CIB2 cation sensitivity and structural transitions. (**a**) In the absence of cations, WT CIB2 is in a molten globule state (left). The addition of a physiological concentration of both Mg^2+^ and Ca^2+^ (middle top panel) triggers the binding of Mg^2+^ in EF3 and EF4. In the presence of physiological Ca^2+^ and absence of Mg^2+^ (middle bottom panel), Ca^2+^ binding is less favorable. When extra (non-physiological) Ca^2+^ concentrations are added, Ca^2+^ replaces Mg^2+^ in EF4 (upper right panel). (**b**) E64D CIB2 is in a molten globule state under physiological conditions (left). A well-defined three-dimensional structure is acquired upon the addition of extra (non-physiological) Mg^2+^ (upper right panel) or Ca^2+^ (bottom right panel), with a more pronounced effect for Ca^2+^. The Wi-Fi logo represents the strength of allosteric interactions between E64 and N121 residues.

**Figure 3 ijms-23-03552-f003:**
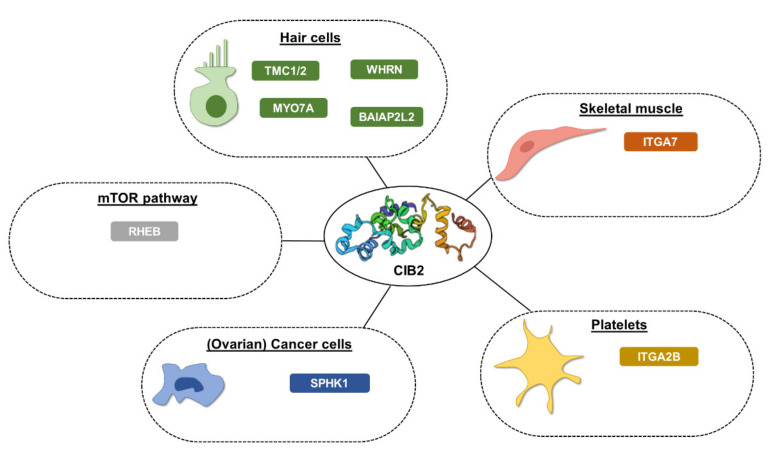
Experimentally determined CIB2 interactors as classified based on cellular type. Direct interactors supported by experimental evidence are listed. WHRN, whirlin; TMC1/2, transmembrane channel-like protein 1/2; MYO7A, unconventional myosin-VIIa; RHEB, GTP-binding protein Rheb; BAIAP2L2, BAI1-associated protein 2-like 2; ITGA7, integrin alpha-7; SPHK1, sphingosine kinase 1; ITGA2B, integrin alpha-IIb.

**Table 1 ijms-23-03552-t001:** Summary of CIB2–integrin interaction studies.

CIB2		Integrin			Method		Ref.
Origin	Tag	Name	Origin	Seq.	Assay	T (°C), Ions	
M.	Yes	αIIb	M.	LVLAMWKAGFFKRNR	ELISA	25 °C	[8]
M.	No	α7B	M.	LAADWHPELGPDGHPVPATA	ITF	20 °C, Ca^2+^	[3]
H.	Yes	αIIb α7B_Mα7B_C	H. M. M.	LVLAMWKVGFFKRNRPPLEEDDEEGQ LVLLLWKLGFFKRA LAADWHPELGPDGHPVPATA	ITF	25 °C, apo, Mg^2+^ and Ca^2+^	[13]
H.	No	α7B_M	H.	LVLLLWKLGFFKRA	ITF	37 °C, Mg^2+^ Ca^2+^	[14]
H.	Yes	α7B_Mα7B_C	H.	LVLLLWKLGFFKRA LAADWHPELGPDGHPVPATA	SPR	25 °C, Mg^2+^ vs. Mg^2+^ and Ca^2+^	[12]

M. = mouse, H. = human. Tag refers to the poly-His tag at the N-terminal. Abbreviations: ITF, intrinsic tryptophan fluorescence; SPR, surface plasmon resonance; ELISA, enzyme-linked immunosorbent assay.

## Data Availability

Not applicable.

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
