# Peer review of "Calcium- and Integrin-Binding Protein 2 (CIB2) in Physiology and Disease: Bright and Dark Sides"

_ijms, 2022, doi:10.3390/ijms23073552_

Round 1
Reviewer 1 Report
In general, this paper is thorough and well-organized.
I had two minor questions that I think the authors should address within the text of the manuscript.
- On page 4 there is discussion that the protein responds with similar intensity to Mg2+ and Ca2+. However, in other proteins, EF-hands tends to bind cooperatively in the presence of Ca2+, and it is not clear the extent to which this might occur in the CIB proteins discussed. I think it would be good to clarify this in the review.
- Can the authors please clarify how NMR revealed allosteric mechanism between EF1 and EF3?
In addition to these questions, it would be beneficial for the authors to complete an additional proofreading of the manuscript for grammatical errors and inconsistencies, some of which I have noted below.
Page 2, Line 50 should read "..as diverse as autophagy..."
Page 2, Line 82 Replace comma after "helix" with dash.
Page 7, Line 277 should read “…was independent of…”
Page 8, Line 287 Change “named” to “called”.
Page 10, Line 393, add comma after (RPE).
Page 10, Line 400, change “associated to”, to “associated with”.
Page 10, Line 409, “functions”.
Page 10, Line 421, “reported”.
Page 11, Line 429, “stiffness”.
Page 11, Line 462, change “poor” to “sparse” or “limited”.
Page 12, Line 488, “demonstrated that CIB2 was”.
Page 12, Line 494, replace “finding” with “who found”, replace “regulates” with “regulated”.
Page 13, “At present, 13 identified mutations in the gene encoding CIB2 have been associated with different forms of hearing impairment, including five non-sense mutation with an ultra-rare frameshift…”.
Page 13, Line 544, “cations, and allosteric effects, or modify…”.
Page 13, Line 578, “capable of binding”.
Page 14, Line 586 “consisting of”.
Page 15, Lines 627-629, remove the semicolon and break this into two sentences.
Page 16, Line 649, “CIB2 could act…”.
Author Response
We are glad that the Reviewer found merit in our Review article and considers it suitable for publication in IJMS following revision.
We thank the Reviewer for helpful comments aimed at improving our manuscript. We have addressed all the points raised by the Reviewer and amended our manuscript accordingly.
-----
- On page 4 there is discussion that the protein responds with similar intensity to Mg2+ and Ca2+. However, in other proteins, EF-hands tends to bind cooperatively in the presence of Ca2+, and it is not clear the extent to which this might occur in the CIB proteins discussed. I think it would be good to clarify this in the review.
We thank the Reviewer for pointing out that, in our original manuscript, we did not mention cooperativity of Ca2+ binding to CIB2. This mechanism was indeed detected by NMR spectroscopy. We have added a sentence in the revised manuscript (lines 121-125), which specifies the presence of positive cooperativity.
- Can the authors please clarify how NMR revealed allosteric mechanism between EF1 and EF3?
In the revised manuscript (lines 191-194) we have added a sentence that should clarify how NMR spectroscopy revealed an allosteric coupling bewteen EF1 and EF3:
"This was clearly demonstrated by 1H-15N HSQC NMR spectra, where the chemical shift of N121 in EF3 and E64 in EF1 were found to show the same pattern upon Ca2+-titration experiments."
In addition to these questions, it would be beneficial for the authors to complete an additional proofreading of the manuscript for grammatical errors and inconsistencies, some of which I have noted below.
Page 2, Line 50 should read "..as diverse as autophagy..."
Page 2, Line 82 Replace comma after "helix" with dash.
Page 7, Line 277 should read “…was independent of…”
Page 8, Line 287 Change “named” to “called”.
Page 10, Line 393, add comma after (RPE).
Page 10, Line 400, change “associated to”, to “associated with”.
Page 10, Line 409, “functions”.
Page 10, Line 421, “reported”.
Page 11, Line 429, “stiffness”.
Page 11, Line 462, change “poor” to “sparse” or “limited”.
Page 12, Line 488, “demonstrated that CIB2 was”.
Page 12, Line 494, replace “finding” with “who found”, replace “regulates” with “regulated”.
Page 13, “At present, 13 identified mutations in the gene encoding CIB2 have been associated with different forms of hearing impairment, including five non-sense mutation with an ultra-rare frameshift…”.
Page 13, Line 544, “cations, and allosteric effects, or modify…”.
Page 13, Line 578, “capable of binding”.
Page 14, Line 586 “consisting of”.
Page 15, Lines 627-629, remove the semicolon and break this into two sentences.
Page 16, Line 649, “CIB2 could act…”.
We are very grateful to this Reviewer for the careful analysis of the text, which revealed several typos. In the revised manuscript we have made all corrections as requested.
Reviewer 2 Report
The review by Dal Cortivo G. and Dell’Orco D. entitled “Calcium- and Integrin-Binding Protein 2 (CIB2) in Physiology and Disease: Bright and Dark Sides” provides a very nice overview on what is known about CIB2 in terms of molecular structures, metal binding properties, interacting proteins and a wide variety of physiological and pathological processes, especially in hearing and related diseases. A detailed description of the structural properties of CIB2 and experimental data for cation sensitivity of CIB2 adds to understanding of the molecular basis of cation sensitivity. The authors have managed to summarize interaction studies of CIB2 with integrins in a table. It is important that the authors listed the methods and the conditions for examining the interactions. This provides valuable information and leads to understanding of the current model for CIB2 in integrin signaling. Further, the authors focus on potential roles of CIB2 in ovarian cancer, infection of HIV, mTORC1 signaling and hearing and deafness. Figures and tables support the main issues of the articles in clear way with exception of Figure 3, which is somehow confusing. I have only minor specific comments:
Figure 1:
In the legend, in lines 103-104, ‘The N-terminal region…’ should be deleted. It should mention in the legend what the two red spheres in the three-dimensional structure are. In the top alignment, it may be help understanding if the EF-hand loops and metal ion-coordinating positions are indicated.
Table 2:
Line 5, ‘c.196T>C” should be ‘c.196C>T’?
Line 6, ‘p.(Glu64Asp9’ should be ‘p.(Glu64Asp)’.
The ethnicities in the table line 3 (Arabic) and line 4 (American) are not match those in the main text, line 536 ‘Nigerian’ and line 538 ‘European’, respectively.
Figure 3:
‘Scheletal muscle’ should be ‘Skeletal muscle’.
Why are Rheb, BAIAP2L2, and TMC2 not listed in the Figure, even though their interactions with CIB2 are supported by co-precipitation? For PCDH15, USH1G, USH1C, and USH2A, experimental evidence should be provided in the main text. In legend, the authors state ‘Only interactors supported by direct experimental evidence are listed’. Does this mean that “Direct interactors supported by experimental evidence are listed’?
Human gene symbols should be italicized with all letters in uppercase, e.g. CIB2 (line 27, etc).
The following abbreviations should be defined: TNS (line 131), OHC (line 576) and IHC (line 577).
Line 216: ‘GSH tag’ replace by: GSH?
Line 429: ‘stfnness’ replace by: stiffness
Author Response
We are glad that the Reviewer found merit in our Review article and considers it suitable for publication in IJMS following revision.
We thank the Reviewer for helpful comments aimed at improving our manuscript. We have addressed all the points raised by the Reviewer and amended our manuscript accordingly.
-----
- Figure 1:
In the legend, in lines 103-104, ‘The N-terminal region…’ should be deleted. It should mention in the legend what the two red spheres in the three-dimensional structure are. In the top alignment, it may be help understanding if the EF-hand loops and metal ion-coordinating positions are indicated.
We thank the reviewer for the suggestion. In the caption of the revised figure, we specified that the red spheres refer to Ca2+ ions. Regarding the upper alignment, we agree that showing EF-hand motifs and calcium-coordinating residues would help the overall understanding. In the revised manuscript, we have modified Figure 1 accordingly, and added this sentence to the legend:
'EF-hand motifs are displayed by colored boxes and residues involved in Ca2+-coordination are indicated by letters referring to the canonical pentagonal bipyramidal geometry on the respective loops. '
- Table 2:
Line 5, ‘c.196T>C” should be ‘c.196C>T’?
Line 6, ‘p.(Glu64Asp9’ should be ‘p.(Glu64Asp)’.
The ethnicities in the table line 3 (Arabic) and line 4 (American) are not match those in the main text, line 536 ‘Nigerian’ and line 538 ‘European’, respectively.
Thank you for highlighting the typos, We have corrected Table 2 according to the suggestions, and checked that ethnicities now match those in the main text.
- Figure 3:
‘Scheletal muscle’ should be ‘Skeletal muscle’.
Why are Rheb, BAIAP2L2, and TMC2 not listed in the Figure, even though their interactions with CIB2 are supported by co-precipitation? For PCDH15, USH1G, USH1C, and USH2A, experimental evidence should be provided in the main text. In legend, the authors state ‘Only interactors supported by direct experimental evidence are listed’. Does this mean that “Direct interactors supported by experimental evidence are listed’?
We thank the Reviewer for highlighting inconsistencies in the originally submitted Figure 3, and for highlighting a typo. Indeed, some interactors (RHEB, BAIAP2L2 and TMC2) were missing, and some others should have not been present, as they are predicted, but not experimentally determined. We have modified Figure 3 accordingly in the revised manuscript. We have also modified figure legend as requested.
- Human gene symbols should be italicized with all letters in uppercase, e.g. CIB2 (line 27, etc).
We italicized gene symbols in the revised manuscript.
- The following abbreviations should be defined: TNS (line 131), OHC (line 576) and IHC (line 577).
Thank you for pointing out that these abbreviations were not defined. We provided definitions in the revised manuscript (lines 144 and 618, respectively).
- Line 216: ‘GSH tag’ replace by: GSH?
We replaced 'tag' with 'moiety' in the revised manuscript (Line 232)
- Line 429: ‘stfnness’ replace by: stiffness
Corrected, thank you!